# Biotransformations Performed by Yeasts on Aromatic Compounds Provided by Hop—A Review

**Stefano Buiatti, Lara Tat, Andrea Natolino**  **and Paolo Passaghe ***

Department of Agricultural, Food, Environmental and Animal Sciences, University of Udine, via Sondrio 2/a, 33100 Udine, Italy; stefano.buiatti@uniud.it (S.B.)
* Correspondence: paolo.passaghe@uniud.it; Tel.: +39-3462365112

**Abstract:** The biodiversity of some *Saccharomyces (S.)* strains for fermentative activity and metabolic capacities is an important research area in brewing technology. Yeast metabolism can render simple beers very elaborate. In this review, we examine much research addressed to the study of how different yeast strains can influence aroma by chemically interacting with specific aromatic compounds (mainly terpenes) from the hop. These reactions are commonly referred to as biotransformations. Exploiting biotransformations to increase the product's aroma and use less hop goes exactly in the direction of higher sustainability of the brewing process, as the hop generally represents the highest part of the raw materials cost, and its reduction allows to diminish its environmental impact.

**Keywords:** biotransformations; aroma; yeasts; fermentation; hop

## 1. Introduction

The increasing popularity of some aromatic beers, such as IPA (India Pale Ale), increased the importance of a better understanding of the chemistry underlying the aromatic characterisation of the product. The composition of the beer aromatic compounds can be regulated not only by modifying the amount, the kind, or the timing of hop addition but also by promoting specific yeast-mediated biotransformation reactions during the fermentation process. The monitoring and/or regulation of these reactions is important not only to produce beers with a desirable aromatic profile but also to ensure production sustainability [1]. Any interventions increasing the transfer of aromatic compounds, thus limiting hop use, will have a direct impact on process efficiency, simultaneously reducing the dependence on a crop that is highly susceptible to damages caused by climate changes [2]. The chemistry underlying beer bitterness caused by hop, i.e., alpha-acid isomerisation into trans- and cis-iso-acids, is well known [3,4]. As a consequence, and above all, thanks to the availability (in different formats) of hops containing well-defined levels of alpha-acids, the bitter taste of the beer can be controlled with good accuracy [5,6]. The chemistry explaining the aroma provided by hop has yet to be completely understood. The "mystery" around the hop aroma (hoppy aroma) of the beer could be caused mainly by the complex composition of the hop essential oil, by chemical/biological conversions that the essential oil constituents can undergo during the fermentation process, and by the fact that the hop aroma arises from additive/synergistic interactions between several volatile compounds [7–11]. Biotransformations are susceptible to different variables [12,13]: the yeast strain, the year and the place of the hop harvest, its variety, and finally, the process conditions (Figure 1).

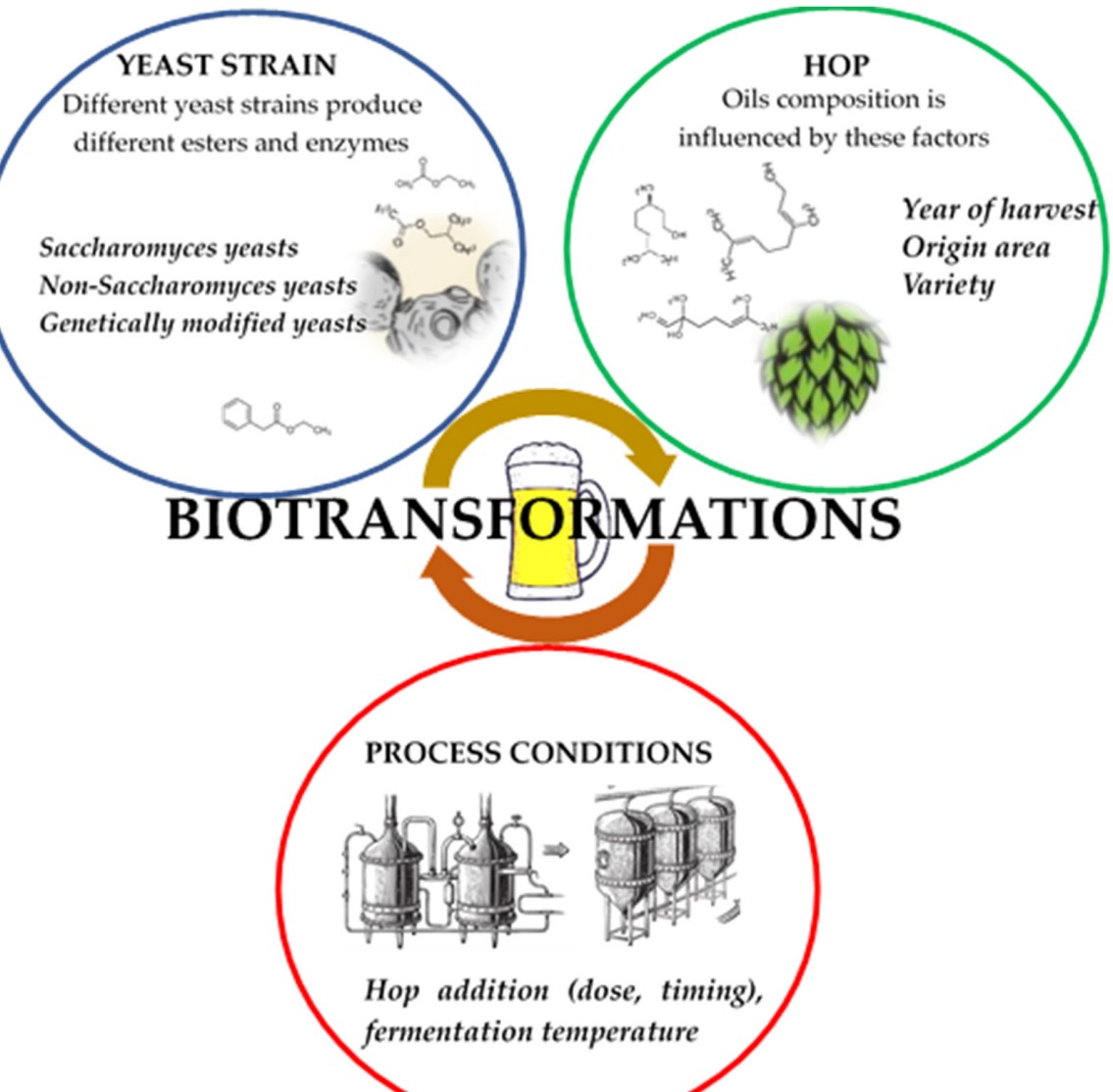

**Figure 1.** The factors that can influence biotransformations of the aroma compounds provided by hops.

## 2. Yeasts Enzymatic Activity for the Release of Aromatic Compound from Precursors

In the hop, monoterpene alcohols can be in a glycosidically bonded form, and their concentration, as well as that of terpenes and terpene alcohols, depends on genetic factors [14–16]. The first study on glycosides and their corresponding volatile aglycones in the hop has been made by [14], and their quantification by [17]. Glycosides are synthesised by an enzyme (glycosyltransferase) that adds an activated sugar (e.g., rhamnose, galactose, xylose, glucuronic acids, etc.) to an aglycone. The beta-glucosidase enzyme activity causes the release of an aromatic terpene (and of a glucose molecule) from a nonaromatic terpenyl glycoside (Figure 2). Some yeast strains are known to have higher levels of enzymatic activity associated with biotransformation, among them beta-glucosidase and beta-lyase [18,19].

**Figure 2.** Beta-glucosidase activity on glucosides and bioconversion of monoterpene alcohols by yeast. Flavour descriptors adapted from the literature; modified from [19].

Both acid and enzymatic hydrolysis of the glycosidic fraction in the beer lead to the release of a series of compounds: aliphatic alcohols (e.g., 6-methyl-5-hepten-2-ol), aromatic compounds (for example, methyl salicylate, benzyl alcohol, phenylethyl alcohol, and vanillin), monoterpene alcohols (for example, cis- and trans- linalool oxide, cis- and trans-8-hydroxylinalool, linalool, alpha-terpineol, and geraniol), and norisoprenoids (e.g., 3-hydroxy-7,8-dihydro-beta-ionol, vomifoliol, dihydrovomifoliol, 3-hydroxy-beta-damascone, and beta-damascone) [18]. Aglycones that have an impact on the beer aroma may also include beta-damascenone (odour threshold 150 ppb); some authors [20] monitored beta-damascenone concentration during wine fermentation using different grape varieties. Beta-damascenone concentration increased during fermentation from low or undetectable levels up to concentrations of several parts per billion. A similar effect was found by [8]. In their study, the evolution of beta-damascenone during wort boiling, fermentation, and in the finished beer has been monitored. Moreover, in this study, an increase in molecules has been observed during fermentation. This increase has been explained by the hydrolysis of glycoside precursors by enzymes provided by yeast cells (exo-1,3-beta-glucanase). According to [21], the increase in citronellol during fermentation can be partially explained by its release from glycoside precursors by the action of exo-1,3-beta-glucanase.

Thiols, too can be found in the hop, in a bonded form and in variable amounts depending on hop variety, the year of harvesting, and the storage time. Some studies suggest, as the biogenesis of various sulphur compounds in the hop, the reaction of terpenes with residual sulphur from fungicidal treatment [22,23]. Polyfunctional thiol concentrations are in the range of ng/L and need particularly sensitive methods for their detection [24,25]. Methods to measure concentrations of cysteinylated and glutathionylated precursors (directly in the hop) have been developed only recently [26,27]. The hop may contain the following precursors: 3-mercaptohexan-1-ol (3MH, also referred to as 3-sulfanylhexan-1-ol, 3SH) and 4-mercapto-4-methylpentan-2-one (4MMP, also referred to as 4-methyl-4-sulfanylpentan-2-one, 4MSP) [26]. For the Cascade hop, concentrations of 6.5 mg/kg for 3MH precursor have been reported, and these are 1000 folds higher than free 3MH concentrations. Moreover, recent research detected (for the first time) the presence of cysteinylated and glutathionylated sulfanylalkyl aldehydes and acetates in the hop and in the grape [27]. Bonded precursors

are attached by beta-lyases provided by yeast with the consequent release of volatile sulphur compounds [28]; these compounds are generally associated with the tropical aroma and have very low perceiving thresholds (Figure 3).

**Figure 3.** In this example, 4-methyl-4-sulfanylpentan-2-one (4-MSP) and cysteine are released from a nonaromatic cysteinylated precursor; modified from [28].

In *S. cerevisiae*, beta-lyase is codified by the IRC7 gene [29,30]. Besides IRC7, beta-lyase codified by the STR3 gene has also been shown to be able to release volatile thiols from 3MH and 4MMP cysteinylated precursors [31]. Gene expression is related to the nitrogen repression process; in the presence of nitrogen sources, such as ammonium, IRC7 is repressed [32]. This aspect highlights the importance of an adequate nutritional strategy in order to maximise the conversion of aromatic precursors. Recently, to select beer yeasts with higher beta-lyase activity, their ability to grow in media containing cysteine as the sole nitrogen source has been exploited [33]. Despite the presence of a wide pool of thiol precursors in the fermentation media, only a small fraction (lower than 1%) is converted into the free form [34,35]. This low conversion could be related to the poor beta-lyase activity in acid media and to the inhibition exerted by polyphenols [36–43]. Heterocyclic terpenes containing sulphur atoms (e.g., myrcene disulphide) can undergo ring opening as a consequence of the reduced activity of the yeast (Figure 4). The release of these aromatic compounds takes place following the cleavage of the C-S bond by beta-lyase [44].

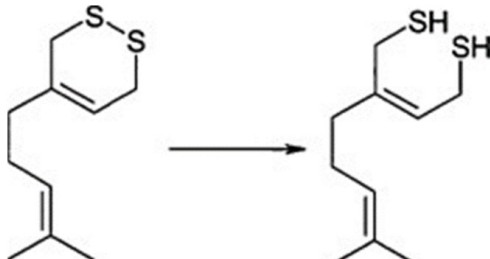

**Figure 4.** Thiol formation via opening of myrcene disulphide heterocyclic ring by the yeast reducing activity, from [44].

During fermentation, a significative reduction in dimethyl trisulphide (DMTS) level and of some higher thioesters takes place. However, some sulphur compounds could survive through ageing and packaging [28]. The optimal conditions to increase the conversion rate of precursors into free thiols still have to be determined.

### 3. The Yeasts

Among *Saccharomyces*, only some species possess a gene coding for beta-glucosidase (Table 1).

**Table 1.** Enzymatic activity associated with biotransformation in *Saccharomyces* and *non-Saccharomyces* yeasts.

| Yeasts | β-Glucosidase Activity | Reference | β-Lyase Activity | Reference |
|---|---|---|---|---|
| *Saccharomyces cerevisiae* | - or +++ | [45] | + or ++ | [35] |
| *Saccharomyces bayanus* | + | [46] | ++ | [47] |
| *Brettanomyces* spp. | ++ or ++++ | [48] | - | [48] |
| *Saccharomyces uvarum* | ++ | [49] | ++ | [47] |
| *Saccharomyces pastorianus* | - | [13] | +++ | [35] |
| *Torulaspora delbrueckii* | - | [50] | ++ | [1,35] |
| *Metschnikowia* spp. | +++ | [50] | ++ | [1] |
| *Lachancea* | + | [50] | ++ | [1] |

(-) No detectable activity; weak activity (+); medium activity (++); strong activity (+++); very strong activity (++++). The relative enzyme activity of different yeasts can only be compared with strains of the same study, as the test conditions were different among the studies.

*S. cerevisiae* yeasts synthesise beta-glucanase regardless of the carbon source, which cleaves the glycosidic bond [51]. Some authors [52,53] proved that the induced biosynthesis of an endogenous exoglucanase in an *S. cerevisiae* strain led to an increase in the release of originally glycosylated volatile compounds in the wort [54,55]. The enzyme is first secreted in the periplasmatic space and then released in the wort during yeast autolysis [56]. The reaction is very specific since the linalool is seldom released, and this could be due to the steric hindrance of the tertiary alcohol itself [57]. However, some researchers [58] observed a release of linalool during wine fermentation performed with *S. cerevisiae* and *S. bayanus*. The hydrolysis of glycosidically bound tertiary alcohols, and then, depends on the strain and the fermentation conditions [13]. The screening according to the exo-beta-glucanase activity of *S.* strains could represent an approach to enhance the taste of the finished product [14]. Glycoside sizes are another variable conditioning biotransformations. If glycoside precursors are characterised by large size, they fail to pass (by simple diffusion) through the plasmatic membrane of living (intact) cells. Genetically improving commercial strains of *S. cerevisiae* by combining their native glycosidase activity with their ability to transport the aromatic precursors from wort into the cell actively represents an extra weapon for the brewer master to produce new beer styles. Furthermore, most worts contain sugars that exert a regulation mechanism known as the catabolite repression effect [59]. Yeast screening for beta-glucosidase activity expressed with high concentrations of these sugars (glucose, sucrose, and fructose) could be very useful. The most interesting and efficient glucosidase activity has been observed in *Brettanomyces custersii* (Table 1). This yeast has been isolated from fermenting lambic [60]. Since Dekkera and *Brettanomyces* species lead to slower fermentations, to reduce the duration of this stage, it can be possible to use a mixed inoculum in which *S. cerevisiae* performs the main fermentation, and *Brettanomyces custersii* is able to release volatile glycosidically bonded aglycones [61,62]. To improve thiol release during fermentation, the interspecific hybridisation between *S. cerevisiae* and *S. uvarum* strains has been used [63]. It appears that *S. uvarum* strains tend to have higher thiol-releasing capacity than *S. cerevisiae* strains [64] (Table 1). Strains that bring specific mutations release higher amounts of thiols [65]. The selection of these strains has been performed through growth on agar plates containing methylamine and proline as the sole nitrogen sources [66]. It has been observed that some lager yeasts can also produce terpene esters such as geraniol and citronellol acetates. Geranyl acetate has a lower perceiving threshold than geraniol and a more characteristic lavender odour [67]. The fact that biotransformations of molecules provided by hop, leading to esters formation, have been observed only in low-fermentation yeasts and not in high-fermentation ones is clearly a reflection of genetic differences between these organisms [68]. A modified *S. cerevisiae* strain, for the gene responsible for the biosynthesis of farnesyl pyrophosphate synthetase (FPPS) enzyme, is able to produce monoterpenoids geraniol, citronellol, and linalool as a consequence of geranyl pyrophosphate (GPP) build-up [69]. Whether linalool is produced as a consequence of an FPPS enzyme "defect" or because of a later biotransformation of

geraniol and citronellol has not been clarified. Using genetically modified yeasts can be a further tool to produce monoterpene derivatives. However, consumers remain critical and concerned about the use of this type of yeast in the brewing sector [13,70].

*Non-Saccharomyces* yeasts for use in fermented beverage production were investigated by different authors. Using *non-Saccharomyces* yeasts for pitching, in sequential or co-inoculation, in bottle conditioning, or many other strategies has presented the possibility of producing high levels of some compounds—namely phenyl ethyl acetate, ethyl hexanoate and ethyl octanoate—that can positively stamp complex fruity, floral and aniseed character to beers [71]. *Metschnikowia* spp. was linked with a great alcohol and ester formation. Among the *non-Saccharomyces* yeasts and the genera *Lachancea* and *Torulaspora* showed a positive influence on the aroma profile of fermented beverages by contributing high amounts of esters, higher alcohols, and other volatile flavour compounds; as esters are volatile flavour compounds which mainly contribute to the aroma of beer, these strains might also be interesting as contributors to beer aroma [72]. With the rise of consumer interest in sour beers, yeasts of the genus *Lachancea* have received increased attention because of their ability to produce both ethanol and lactic acid. *L. thermotolerans* is the most studied species of the genus, with a focus on its ability to acidify beer and impact both the aroma and flavour of the final product [73]. There has also been extensive research on the use of *T. delbrueckii* to introduce complexity and desirable aromas and flavours to beer [74]. Some authors assume that *T. delbrueckii* is also one of the main ones responsible for the fermentation of Bavarian wheat beers [75]. These microorganisms can undergo high osmotic pressure conditions (highly osmotolerant), grow well at low temperatures (cryotolerants), and curiously require oxygen to ferment [76]. *T. delbrueckii* can grow—even with an increase in the lag phase—in the presence of up to 90 ppm isoα-acids in the medium, a concentration that correlates to highly hopped beer styles [77]. King and Dickinson [78] reported that *T. delbrueckii* is able to transform hop terpenoids and significantly influences the aroma profile of the beer produced. *T. delbrueckii* strains have been considered for use in the production of low-alcohol beers by Canonico et al. [79], who screened different strains and observed generally low ethanol production. The production of low-ethanol beers using *non-Saccharomyces* yeasts is an increasingly popular method as it avoids the loss of aromas (mainly of higher alcohols and esters) and the body that occurs when alcohol is mechanically removed from the beer [80].

## 4. The Hops

The hop (*Humulus lupulus*) is one of the plants with the most complex essential oil composition. To date, more than 200 compounds have been identified [81]. An analysis of the impact of climate change on hop oil composition (little water and stress from high temperature) should be performed, which will result in the culture of new strains, with all the consequent implications for the brewing sector [82]. Structures of terpenes provided by hop and relevant for beer aroma characterization are shown in Figure 5 [83,84].

Terpenes are a class of biosynthesised compounds from the mevalonic acid pathway. Terpenes can also be produced by an alternative (nonmevalonic) biosynthetic pathway starting from triosephosphates that is called 1-deoxy-D-xylulose-5-phosphate pathway. The two pathways converge in the production of two activated molecules showing the same carbon (isoprene) backbone: isopentenyl diphosphate and dimethylallyl diphosphate. Terpenes originate from these two molecules. Terpenes are widespread in nature and include monoterpenes (C10), sesquiterpenes (C15), diterpenes (C20), triterpenes (C30), carotenoids, sterols, phytols, and quinones [85]. Monoterpenes are compounds with marked sensory qualities, and they are found in the essential oils of hops [83]. Terpenes aromas vary widely and include the following notes: floral, fruity, menthol, and peppery. Isomers of a given terpene can have different aromas. For example, geraniol has a rose, lime, and flower smell, while the cis isomer (nerol) has a fresh, "green" smell [62,86]. Terpene hydrocarbons provided by hop (e.g., beta-myrcene, alpha-humulene, and beta-caryophyllene) adsorb on yeast cell walls and are removed during flocculation or filtration [87–89]. Studies have

also been performed on oxygenated derivatives of monoterpenes forming during hop storage [90,91]. It would be interesting to determine the fate of these compounds in the presence of yeast. Oxygenated derivatives of main mono and sesquiterpenes derivatives (e.g., humulene, caryophyllene epoxides, and alcohols) have a higher probability than non-oxygenated hydrocarbons to remain in the finished beer and then contribute to its "hopped" aroma [68]. Linalool is present in most, if not all, hops (odour threshold in R-form of 2.2 ppb). Beta-citronellol is present in all hops, though in traces. Geraniol is present in different amounts depending on the variety: for example, European hop varieties (Hallertauer Tradition, Hallertauer Magnum, Saaz, etc.) contain very small amounts of geraniol, while USA hop varieties (Apollo, Amarillo, Bravo, Cascade, Citra, Mosaic etc.) often contain big amounts of this terpene alcohol [92]. Using copper in hop treatment affects the content of certain volatile sulphurated compounds that are present in the beer [93,94]. In some hops varieties (Comet, Amarillo, Polaris, Summit, and Vic Secret), geraniol is present in a bonded form (glycoside). It is then released more slowly and is not always transformed into other compounds [92]. Geranyl esters, widely present in several hop varieties (e.g., Cascade), can be hydrolysed into geraniol during fermentation through the activity of acetate esterase. Using hop varieties containing high concentrations of geranyl acetate, beers are produced with a low level of geranyl acetate but a high level of geraniol. This suggested that these esters are hydrolysed during fermentation [7].

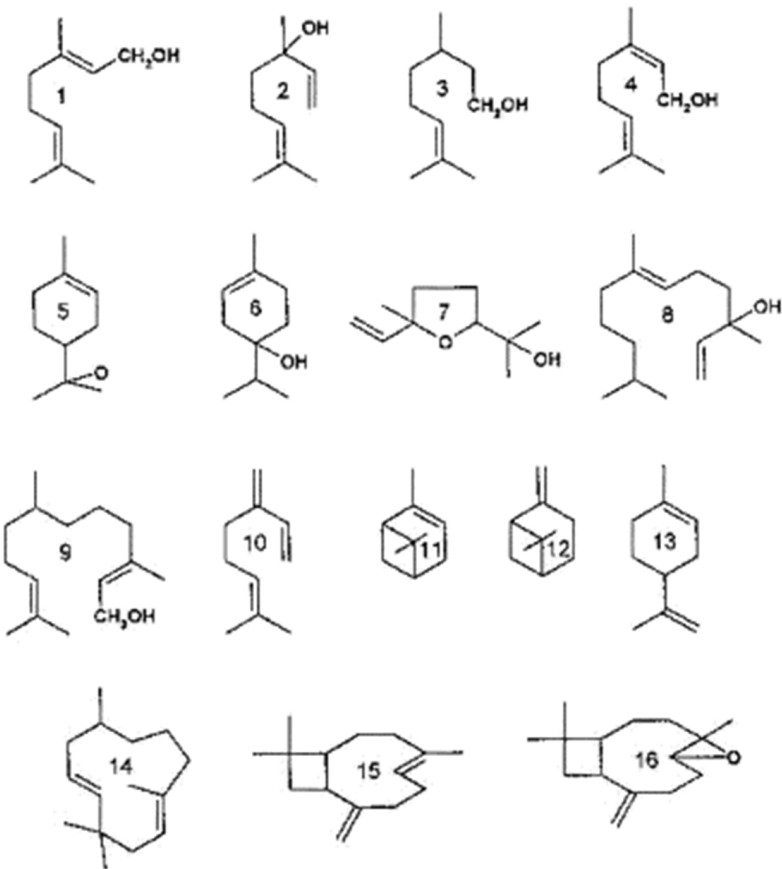

**Figure 5.** The structures of terpenoids relevant to brewing and their aromas. 1: geraniol (floral, rose-like, citrus); 2: linalool (floral, fresh, coriander); 3: citronellol (sweet, rose-like, citrus); 4: nerol (floral, fresh, green); 5: K-terpineol (lilac); 6: terpinen-4-ol (spicy, medicinal); 7: linalool oxide (herbaceous, medicinal); 8: nerolidol (floral); 9: farnesol (floral); 10: L-myrcene (medicinal); 11: K-pinene (pine); 12: L-pinene (pine); 13: limonene (sweet, spicy, citrus); 14: K-humulene (medicinal); 15: L-caryophyllene (herbaceous); 16: L-caryophyllene oxide (spicy). From [83].

## 5. Biotransformations during Beer Production

As known, hop is added at particular stages during boiling to provide bitterness; the addition is made at the beginning of the boiling stage, while to characterise the aroma, the addition takes place at the end of the boiling stage (late hopping), and/or during the lagering stage (dry hopping) [95]. The dry hopping process is particularly expensive for the use of water and raw materials. When the hop is removed, a great beer loss takes place [96]. Moreover, hop reuse is compromised when using "static" dry hopping. In "dynamic" dry hopping, the hop is added when the fermentable activity is ended [97]. The hop dosing timing (with or without the yeast) affects the aromatic profile of the beer [1]. The effect of time and the boiling system on the concentration of some terpenes and terpenoids has been studied [98]. In hopped beers with the late- and dry-hopping techniques, linalool was present above its aromatic threshold. Thus, it has been proposed as an analytical marker both for intensity and the quality of the "hopped" aroma of these kinds of products [99,100]. A late addition ("dynamic" dry hopping) of a hop with a high concentration of geraniol is a production approach leading to the control of citrus characteristics of the finished beer [101].

The temperature during fermentation affects thiol release from their conjugated precursors. However, the temperature effect varies according to the kind of thiol: 3MH and 3MP (3-mercaptopenthanol, also referred to as 3-sulphanyl pentan-1-ol, 3SP) are mostly released in the range 18–24 °C, 3S4MP (3-sulphanyl-4-methylpentan-1-ol) at 28 °C. Post-fermentative ageing, performed at 4 °C for up to 5 days, leads to a constant increase in thiol concentration [102]. The addition of exogenous enzymes during fermentation (e.g., cystathionine beta-lyase and apotryptophanase) was also evaluated in order to increase the release of thiols from cysteinylated precursors. However, significant differences have not been observed compared to non-added media [27]. Some enzymes (Aromazyme, Rapidase, or Sumyzime) are already available on the market and are used to maximise the release of glycosylated monoterpene alcohols [92]. Encouraging data have been collected on the release of some bonded monoterpene alcohols and on alpha-L-arabinofuranosidase and alpha-L-ramnosidase [16].

## 6. Other Reactions on Molecules Provided by Hop

To define chemisms leading to the formation of certain molecules, the "labelling" technique of precursors/intermediates is often used. "To label" a molecule means to replace a constituent atom with its isotope. The "labelled" precursor or intermediate, metabolised once introduced in the biological cycle of yeast cells, is isolated, purified, and analysed to assess its isotope content. If the isotope incorporation occurred, we could reasonably think about a relation between the precursor and the metabolite. As a consequence, it can be suggested a chemical way for that transformation [103]. Some authors [103] exploited the "labelling" technique (with deuterium) of the water in the fermentation wort in order to define the biosynthesis (during wine fermentation) of (cis) rose oxide. The collected results highlighted that enzymes provided by yeast reduce the 3,7-dimethylocta-2,5-dien-1,7-diol (geranyl diol I) precursor and produce 3,7-dimethyl-5-octen-1,7-diol (citronellyl diol I), which gives rise to the rose oxide with a cyclisation reaction. The rose oxide present in the hop essential oil and in the finished beer was detected [104,105]. Aglycones released in the wort can be transformed into other aromatic molecules during the lagering stage by yeast-derived enzymes. The backbone modifications through oxidations, reductions, isomerisations, conjugations, ring closures, hydrations, and dehydrations produce thousands of different terpenes [78] (Figure 6).

1. Reduction of geraniol to citronellol (except *T. delbrueckii*)
2. Isomerisation of geraniol to nerol
3. Isomerisation of nerol to linalool
4. Isomerisation of linalool to α-terpineol
5. Isomersiation of nerol to α-terpineol
6. Hydration of α-terpineol to terpin hydrate
7. Isomerisation of nerol to geraniol (except *S. cerevisiae*)

**Figure 6.** Scheme showing the monoterpenoid biotransformation reactions catalysed by *S. cerevisiae*, *Torulspora delbrueckii*, and *Kluyveromyces lactis*; adapted from [78].

Enzymes provided by yeast cells act in regiospecificity (in the case of several functional groups, the reaction takes place in only one specific site) and stereoselectivity (the enzyme leads to the formation of only one enantiomer) conditions. For example, geraniol reduction is stereospecific and leads to 100% of the enantiomeric R form of citronellol [13,106]. In the 2000s, a chemism was proposed for monoterpene alcohol biotransformations by the yeast based on the results collected by fermentation of model systems containing monoterpene alcohols [92]. The composition of models (made of fermentable syrup and added with terpenes and monoterpene alcohols) inoculated with a high-fermentation yeast in one case and a low-fermentation yeast in the other has been monitored. Since catabolite (carbon) repression is an important regulator for yeast metabolism, the idea at the base of this study was to investigate hop terpenoids transformation in models with high sugar concentrations, getting closer (as for composition) to the traditional beer worts. The nitrogen source concentration is another important regulatory factor for yeast. However, carbon level was shown to have the greatest effect on yeast metabolism compared to changes in nitrogen concentration [68,107]. *S. cerevisiae* is able to biotransform geraniol and nerol into linalool, isomerise cis-trans nerol to give geraniol, and perform nerol and linalool cyclisation producing alpha-terpineol (Figure 2). Yeast enzymes, through alpha-terpineol hydroxylation, produce terpin that can be further hydrated probably by spontaneous reactions. Nerol conversion into alpha-terpineol takes place more rapidly compared with the conversion of linalool into alpha-terpineol [68,95].

It has been proposed that the yeast-derived OYE "Old Yellow Enzyme" enzyme (dehydrogenase) is responsible for the catalysis of geraniol reduction into beta-citronellol (Figure 2); OYE has been discovered in the 1930s and used to demonstrate (for the first time) the need of a cofactor (nicotinamide adenine dinucleotide phosphate, NADP) for the catalysis of certain reactions by enzymes. Geraniol is mainly converted 2–4 days after the beginning of the fermentation [108,109]. Carbonyl compounds can be reduced into alcohol by yeast [7,110]. For example, methyl ketones are partially reduced into the corresponding secondary alcohols [87]. In the hop essential oil, saturated and unsaturated aldehydes (geranial, neral, and citronellal) and several ketones have been detected [111]. Dehydrogenase and reductase are the key enzymes that catalyse the reduction of a carbonyl group into a hydroxyl group. Alcohol dehydrogenase and enoate reductase explain the reduction in

carbonyls and "activated" (with an electron attractor substituent, for example, carbonyl or carboxyl) carbon-carbon double bonds, respectively. Both enzymes require the oxidation of a nicotinamide adenine dinucleotide diphosphate coenzyme; the reduced form (NADPH) is regenerated through molecules that behave as a hydrogen source, such as alcohol and glucose [112]. Linalool, which has the lowest perceiving threshold among terpene alcohols, could act as a key element in additive effects with citronellol and geraniol [21]. From the sensory chemistry perspective, the existence of an additive effect between linalool, geraniol, and beta-citronellol has been shown. The citrus odour is the result of the coexistence of these three monoterpene alcohols [113]. The occurrence of small amounts of geraniol and nerol in tests performed on models added with linalool has made it possible to emphasise that some biotransformation reactions are reversible. Specifically, when the wort changes its composition, i.e., when a lower nutrient level, more ethanol, and no oxygen residues are present, and with depleted sterols intracellular reserves, yeast biotransformations can take place in the opposite direction [95]. Tests performed on models added with terpenes showed that the terpenes themselves did not influence the fermentation process, and this is a sign that they are not toxic for the yeasts. However, it has been observed that fermentations of models added with terpenes were less vigorous than those of not-added models [68]. Esters can undergo (during fermentation) both hydrolysis and transesterification [114,115]. The biogenesis of citronellyl acetate has not yet been clarified, i.e., whether it is formed by citronellol esterification, geranyl acetate reduction, or both chemisms. In some cases, methyl esters (e.g., methyl geranate) are not hydrolysed, so they are detectable in the beer [111].

### 7. Work in Progress and Future Perspectives

As biotransformations are influenced by several variables, our working group is performing a comparative study on beers produced with three hop varieties (added in different moments of the process) and the same selected yeast (with a high glucosidase activity). Specifically, two noble hops (high levels of aromatic notes) and a traditional variety have been used. The aim of this work is to understand the evolution of aromatic hop-derived compounds during fermentation and to maximise the efficiency of the brewing process. A deeper knowledge in this field can contribute to the promotion of traditional hops, raising the range of flavours that they can produce and exploiting the wide genetic diversity of the yeast species available. The objective is also to increase our ability to develop products with tailor-made aromatic profiles in an efficient and sustainable way.

**Author Contributions:** S.B., writing—original draft preparation and proofreading the manuscript; L.T., proofreading the manuscript; A.N., proofreading the manuscript; P.P., writing—original draft preparation. All authors have read and agreed to the published version of the manuscript.

**Funding:** This research received no external funding.

**Institutional Review Board Statement:** Not applicable.

**Informed Consent Statement:** Not applicable.

**Data Availability Statement:** Data sharing not applicable. No new data were created or analyzed in this study. Data sharing is not applicable to this article.

**Conflicts of Interest:** The authors declare no conflict of interest.

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
