# Peer review of "Biotransformations Performed by Yeasts on Aromatic Compounds Provided by Hop—A Review"

_fermentation, doi:10.3390/fermentation9040327_

Round 1
Reviewer 1 Report
Dear Authors:
The review is well written and documented, with a topic that may be of use to researchers concerned with yeast biochemistry. However, some minor changes and clarifications are needed to improve the manuscript to be published in Fermentation 2023:
Figure 1 is too basic for a review, in my opinion it should detail more information or be accompanied by graphs or illustrations: it looks more like a table, in which case more data in it would be appreciated.
line 100 and others. Sacch. cerevisiae should be modified: the most commonly accepted abbreviation is S. cerevisiae (indicate the first time the full name and the next time use this abbreviation).
line 122: the authors should argue and provide a citation for this sentence "Only some Saccharomyces cerevisiae yeasts possess a gene coding for beta-glucosidase" since other non-Saccharomyces species have been shown to have such activity.
More references on aromatic composition related to non-Saccharomyces yeasts are missing throughout the document. Please include discussions on some species such as Metschnikowia, Lachancea, Torulaspora... with their respective international citations.
Congratulations for your work in the advancement of knowledge.
Reviewer 2 Report
The review is interesting and it has references necessary on the topic, but in the section of yeast the authors must add a table.
The table should have a column of type of yeast and other column of type of reaction.
